# Biological Activity of *fac*-[Re(CO)_3_(phen)(aspirin)], *fac*-[Re(CO)_3_(phen)(indomethacin)] and Their Original Counterparts against Ishikawa and HEC-1A Endometrial Cancer Cells

**DOI:** 10.3390/ijms231911568

**Published:** 2022-09-30

**Authors:** Olga Kuźmycz, Aleksandra Kowalczyk, Paweł Stączek

**Affiliations:** Department of Molecular Microbiology, Institute of Microbiology, Biotechnology and Immunology, Faculty of Biology and Environmental Protection, University of Lodz, 12/16 Banacha, 90-237 Lodz, Poland

**Keywords:** endometrial cancer, NSAIDs, rhenium compounds

## Abstract

Nonsteroidal anti-inflammatory drugs (NSAIDs) are inhibitors of cyclooxygenase enzyme (COX) and were found to have positive effects in reducing the risk of developing gynecological cancers. However, long-term administration of NSAIDs carries the risk of various side effects, including those in the digestive and circulatory systems. Therefore, there is a constant need to develop new NSAID derivatives. In this work, we investigated rhenium NSAIDs, comparing their effects on endometrial cancer cells with original NSAIDs, demonstrating the high activity of aspirin and indomethacin derivatives. The cytotoxic activity of rhenium derivatives against the Ishikawa and HEC-1A cancer cell lines was higher than that of the original NSAIDs. The IC_50_ after 24-h incubation of Ishikawa and HEC-1A were 188.06 µM and 394.06 µM for rhenium aspirin and 228.6 µM and 1459.3 µM for rhenium indomethacin, respectively. At the same time, IC_50_ of aspirin and indomethacin were 10,024.42 µM and 3295.3 µM for Ishikawa, and 27,255.8 µM and 5489.3 µM for HEC-1A, respectively. Moreover, these derivatives were found to inhibit the proliferation of both cell lines in a time- and state-dependent manner. The Ishikawa cell proliferation was strongly inhibited by rhenium aspirin and rhenium indomethacin after 72-h incubation (*** = *p* < 0.001), while the HEC-1A proliferation was inhibited by the same agents already after 24-h incubation (*** = *p* < 0.001). Furthermore, the ROS level in the mitochondria of the tested cells generated in the presence of rhenium derivatives was higher than the original NSAIDs. That was associated with rhenium indomethacin exclusively, which had a significant effect (*** = *p* < 0.001) on both Ishikawa and HEC-1A cancer cells. Rhenium aspirin had a significant effect (*** = *p* < 0.001) on the mitochondrial ROS level of Ishikawa cells only. Overall, the research revealed a high potential of the rhenium derivatives of aspirin and indomethacin against endometrial cancer cells compared with the original NSAIDs.

## 1. Introduction

Nonsteroidal anti-inflammatory drugs (NSAIDs) are the most commonly used drugs worldwide. Since their first use in the 1960s, the chemical structure and the principal objective of their usage was modified and changed. Along with their intended application as anti-inflammatory agents, these drugs are currently administered as analgesics, antipyretics, and antiplatelet-aggregation agents [1]. The well-known activity of NSAIDs involves the inhibition of isoforms of the cyclooxygenase (COX) enzyme-COX-1, COX-2, or both, depending on the specific drug. In particular, those NSAIDs capable of inhibiting COX-2 (alone or together with COX-1) are of particular interest in the context of cancer prevention and supportive treatment, as this enzyme has been found to be an inducible isoform associated with inflammation and cancer proliferation [2]. Recently, several research groups have focused on the role of inflammatory processes in the development of gynaecologic cancers [3,4,5], as well as the reduction of the risk of developing these types of cancer as a result of long-term NSAIDs administration. The consumption of aspirin in low doses has already been recommended for the prevention of colorectal cancer and cardiovascular diseases. It has also been suggested to be helpful in the case of lung, breast, and prostate cancers [6,7,8,9]. In addition to the anti-inflammatory effect, the cytostatic activity of NSAIDs was noticed. Chiu et al. [10] reported the inhibition of melanoma cell (A375) proliferation after celecoxib and indomethacin treatment. Marinov et al. [11] indicated the high potential of diclofenac as a cytostatic agent against breast (MCF-7) and colorectal (HT-29) cancer cells but not cervical cancer cells (HeLa). Moreover, studies by Buzharevski et al. [12] performed with carboranyl derivatives of rofecoxib against melanoma (A375, 518A2, B16, B16F10) and colon (HCT116, SW480, SW620, CT25CL26) cancer cells highlighted the potential perspectives of modified NSAIDs as anticancer agents with cytostatic properties. Here, the observed activities of agents imply that the anticancer potential of NSAIDs may not be explained only by their COX-inhibitory activities. Still, the major clinical problem with NSAID consumption in anticancer therapy is gastrointestinal complications. The side effects of NSAIDs (especially those that inhibit COX-1) can lead to gastric/duodenal bleeding and ulceration, so modifications to these commonly used drugs have been necessary. Some of these modifications, such as substitution with nitric oxide (NO), hydrogen sulfide (H_2_S), lipids, and metal complexation, were shown to give a broad spectrum of biological activities. Furthermore, the complexation of NSAIDs with certain metals, such as copper or zinc, reduced the effective dose of the “native” drug [13,14,15]. Complexes of rhenium with aspirin, naproxen, indomethacin, and ibuprofen, were synthesized by Skiba et al. and subsequently tested by our group on HeLa and L929 cell lines [16]. The key aspect of the antitumor activity of these complexes was not the increased inhibition of the COX enzyme but the broadening of the spectrum of activity of the rhenium derivatives of the NSAIDs tested. Indeed, rhenium-aspirin was found to exhibit higher cytotoxicity against cancerous HeLa cells compared with non-cancerous L929 cells. It was also noted that the mechanism inducing cancer cell death was based on an increase in reactive oxygen species (ROS) levels, which was correlated with a cell cycle arrest in sub-G1 and S phases [16]. Here, we report our findings using an in vitro model of the endometrial cancer cell lines Ishikawa and HEC-1-A. Our studies focused on the cytotoxic effects of rhenium-modified NSAIDs, as well as the effects of metal complementation with these agents. 

## 2. Results

### 2.1. Effects of Aspirin, Indomethacin and Their Rhenium Derivatives on Endometrial Cancer Cell Viability

The tested compounds, i.e., aspirin, indomethacin, naproxen, ibuprofen, and their rhenium derivatives, were applied at selected concentrations against Ishikawa and HEC-1A cells and their effects on cell viability were evaluated by the MTT assay. Only two rhenium compounds–rhenium aspirin (**375**) and rhenium-indomethacin (**377**), and their corresponding NSAIDs, aspirin (**375a**) and indomethacin (**377a**), showed an effect on endometrial cancer cell viability. The examination was performed using the same methodology as described in previous studies [16] conducted by our research group.

The IC_50_ values were determined with a tetrazolium (MTT) assay, and the results are summarized in Table 1 for the Ishikawa cell line and Table 2 for the HEC-1A cell line. After 24-h incubation, the cytotoxic activity of rhenium derivatives **375** and **377** against Ishikawa and HEC-1A cancer cell lines was significantly higher than that of **375a** and **377a**, indicating that they were more effective than the corresponding original NSAIDs. IC_50_ for **375** was 188.06 µM and 394.06 µM for Ishikawa and HEC-1A cancer cells, respectively, whereas the IC_50_ of **375a** was several folds higher for both cancer cell lines, reaching 10,024.42 µM and 27,255.8 µM, respectively. The IC_50_ values of **377** for Ishikawa and HEC-1A were 228.6 µM and 1459.3 µM, respectively, while the IC_50_ values of **377a** for both cancer cell lines were 3295.3 µM and 5489.3 µM, respectively. The cytotoxic effect of the reference agent CisPt, was also studied under the same conditions. The IC_50_ for Ishikawa and HEC-1A was 13.57 µM and 20.05 µM, respectively. As the results showed, cells of the Ishikawa line were more susceptible to the compounds during the first 24-h incubation period than cells of the HEC-1A line, which could be related to the pre- and postmenopausal nature of the cells.

After a 72-h incubation period, the enhancement of the cytotoxic effect of both rhenium derivatives on Ishikawa and HEC-1A cancer cells was observed. IC_50_ for **375** in Ishikawa and HEC-1A was 146.3 µM and 112.4 µM, respectively, while the initial NSAID IC_50_ value of **375a** was 17,395.6 µM for Ishikawa and 17,168 µM for HEC-1A cell line. The IC_50_ value of **377** was 107.7 µM for Ishikawa and 138.5 µM for HEC-1A cells, whereas the IC_50_ value of **377a** was 4737.4 µM and 6375.2 µM, respectively. The IC_50_ values of CisPt for both Ishikawa and HEC-1A cancer cell lines were 41.03 µM and 10.27 µM, respectively. After 72-h incubation with the studied agents, the discrepancy between IC_50_ values was lower than that between IC_50_ values after 24-h incubation. Moreover, the inhibitory effect of the agents against HEC-1A cancer cells was increased.

### 2.2. Inhibition of Endometrial Cancer Cell Proliferation by Aspirin, Indomethacin, and Their Rhenium Derivatives

Due to the previous results of the MTT test, further research was continued by using **375**, **375a**, **377**, and **377a**. To evaluate the effects of NSAIDs and their derivatives on the proliferation of Ishikawa and HEC-1A cells, a BrdU assay was performed, which involves the incorporation of BrdU into newly synthesized DNA followed by its detection. Ishikawa and HEC-1A cells were incubated with each compound for 24 and 72 h.

The changes in the proliferation status of Ishikawa endometrial cancer cells were examined for each of the tested compounds at IC_50_ (Figure 1), ½ IC_50_, and ¼ IC_50_ (Table 3) after 24- and 72-h incubation. After 24-h incubation with **375**, **375a**, **377**, and **377a**, only **377** and **377a** showed a significant antiproliferative effect on the tested cells. However, the rhenium derivative **377** inhibited proliferation less (* = *p* < 0.05) than the original NSAID **377a** (*** = *p* < 0.001). Differences in proliferation status of the tested cells were also observed in the case of ½ and ¼ IC_50_. Here, both **377** and **377a** showed higher significance (*** = *p* < 0.001) in inhibiting cell proliferation, compared with **375** and **375a**, where only the effect of **375a** was significant (* = *p* < 0.05). The proliferation inhibitory effect of the reference drug CisPt was tested after 24-h incubation under the same test conditions (* = *p* < 0.05). All tested compounds significantly inhibited the proliferation of Ishikawa cells after 72-h incubation. Only **375a** had a lower proliferation inhibitory effect on cells in IC_50_ (** = *p* < 0.01), compared with the other compounds tested (*** = *p* < 0.001).

The proliferation status of HEC-1A endometrial cancer cells was studied under the same conditions as the Ishikawa cells described above. Studies were performed using the tested compounds at their IC_50_ (Figure 2), ½ IC_50_, and ¼ IC_50_ (Table 4) after 24- and 72-h incubation. After 24-h incubation, only 1 of the tested compounds revealed no significant antiproliferative effect on the studied cells, **375a**, while the other compounds showed high inhibitory activity (*** = *p* < 0.001). This effect was not observed at the lower concentrations of the compounds, where only two, **375** at ½ IC_50_ and **375a** at ¼ IC_50_, showed significant inhibition of cell proliferation (* = *p* < 0.05). The reference drug CisPt was tested after 24-h incubation under the same assay conditions. However, no significant effect on cells was observed. The lack of proliferation inhibitory effect of CisPt on HEC-1A after 24-h incubation could be related to the short incubation time with the drug. Proliferation inhibition by the tested derivatives was enhanced after 72-h incubation (*** = *p* < 0.001), and all agents had a similar effect on HEC-1A cells.

### 2.3. The Effect of Aspirin, Indomethacin and Their Rhenium Derivatives on ROS Level in the Studied Endometrial Cancer Cells

#### 2.3.1. Cytosolic ROS Level Measurement

The ROS are well-known for their signaling capabilities and can trigger oxidative stress-induced cancer cell death. On the other hand, cancer cells can maintain ROS homeostasis and escape the initiation of cell death by the increase of their antioxidant properties [17]. Thus, new drugs that strongly induce the formation of ROS to overcome this defense may prove effective in fighting cancer cells. Therefore, the tested compounds were analyzed in this respect against Ishikawa and HEC-1A cell lines using H_2_DCF-DA dye (Thermo Fisher, Warsaw, Poland).

The measurements in Ishikawa cells (Figure 3) revealed the high ROS inducing activity of aspirin rhenium derivative **375** at ½ IC_50_ (*** = *p* < 0.001), and IC_50_ (* = *p* < 0.05), as well as the native NSAID **375a** at ½ IC_50_ (*** = *p* < 0.001) and IC_50_ (** = *p* < 0.01). In Ishikawa cells, **377** was found to be less effective in generating ROS compared to the native NSAID, with **377a** at ½ IC_50_ and IC_50_ being significant (*** = *p* < 0.001 and * = *p* < 0.05, respectively). In the case of the HEC-1A cell line (Figure 4), the level of ROS produced by the rhenium compounds was significant for both **375** and **377** (*** = *p* < 0.001) but not for corresponding NSAIDs. In addition, both rhenium derivatives induced increased ROS levels at ½ IC_50_ concentrations (*** = *p* < 0.001). However, the activity of the drug was not constant at all measured time points (0.5, 1 or 3 h after incubation), indicating a time-dependent effect of the NSAIDs and rhenium-NSAIDs on ROS induction. In addition, lower drug doses (½ IC_50_) were found to initiate the production of ROS more rapidly, possibly related to the high solubility of the compounds in the culture medium, leading to immediate uptake of the drugs across the cell membrane.

#### 2.3.2. Mitochondrial ROS Level Measurement

It has been found that about 90% of cellular reactive oxygen species is produced by mitochondria. Disruption of this overproduction leads to oxidative stress, which in turn leads to oxidative damage. This damage affects cellular components such as lipids, DNA, and proteins, which is unfavorable for cells [18]. However, the accumulation of rhenium NSAIDs in mitochondria was reported in a previous study [16] and could possibly be considered as an anticancer mechanism of rhenium NSAIDs action. Therefore, mitochondria-specific ROS production was also investigated in endometrial cancer cells.

The mitochondrial ROS changes were observed only after 2-h incubation with studied agents, by using MitoSOX Red dye (Invitrogen, Waltham, MA, USA). These results showed that the rhenium NSAID derivatives produced a significant amount of mitochondrial ROS compared to the original NSAIDs. In addition, differences were found between **375** and **377**. In both Ishikawa (Figure 5a) and HEC-1A cells (Figure 5b), **377** generated more mitochondrial ROS than **375** when these drugs were used at their IC_50_ values. At the same time, no correlation was observed between mitochondrial and cytosolic ROS levels, which is probably related to a different nature of ROS and incubation time. H_2_O_2_ is used as a secondary messenger to coordinate the oxidative metabolism of cells. Levels of H_2_O_2_ are regulated through its production and degradation, where mitochondria serve as “stabilizing devices” and buffer cellular H_2_O_2_ levels. The stabilization of hydrogen peroxide is related to the high concentration of antioxidant defense enzymes in the matrix [19,20]. These properties of H_2_O_2_ cleavage by mitochondria can explain the lack of significant mitochondrial ROS production initiated by 1% H_2_O_2_.

## 3. Discussion

Positive anticancer effects of NSAIDs, including specific action against endometrial cancer, have already been studied [21]. It was found that the positive effect of low-dose aspirin and its synergism with some other NSAIDs were not only associated with good patient survival outcomes but also had an anti-proliferative effect on cancer cells [22,23,24]. These effects were also previously noticed in vitro for Ishikawa and HEC-1A endometrial cancer cells. Studies performed by Gao et al. [25] determined the effects of aspirin, indomethacin, and chosen cyclooxygenase (COX-2)–selective inhibitor-NS398. All three NSAIDs significantly inhibited the proliferation of Ishikawa and HEC-1A cancer cells in a dose- and time-dependent manner. In addition, aspirin and indomethacin were involved in cellular apoptosis initiation through a cytochrome *c*-related pathway. Other studies have shown the inhibition of growth, apoptosis induction, and reduced Bcl-2 expression of Ishikawa cells about 21–88% after aspirin treatment [26,27]. Indomethacin was found to be related to the up-regulation of the PTEN tumor suppressor, which affected the HEC-1B cancer cell line [28]. A selective COX-2 inhibitor-celecoxib, reduced HEC-1A and HEC-1B cell lines growth and assisted in the inhibition of tumor cell proliferation [29]. However, none of these studies used modified NSAIDs. Structural and chemical modifications of NSAIDs may not only enhance the antitumor activity but also reduce the risk of side effects which are common during regular NSAIDs intake [30,31]. For this reason, new NSAID derivatives have been developed and synthesized, such as: NO-NSAIDs (hybrid nitrate NSAIDs), cGMP PDE (cyclic guanosine monophosphate phosphodiesterase), AChE (acetylcholinesterase inhibitors), phospho-NSAIDs, TEMPO (2,2,6,6-tetramethyl-1-piperidinyloxy), TEMPOL (4-hydroxy-TEMPO) and HS-NSAIDs (hydrogen sulfide NSAIDs) [23,24,25,32,33,34]. However, the study omitted endometrial cancer cells, which are known to overexpress COX-2 [35,36]. Because of metal complexation, NSAIDs could also get new anticancer qualities. Some zinc complexes of naproxen had shown high activity against bacterial strains while copper-aspirin complexation had served as superoxide dismutase mimetics and had high cytotoxic activity against selected human cancer cells. These agents were not associated with COX inhibition properties but were involved in DNA binding and cleavage [37,38].

The results of the cell viability assay (MTT) showed the activity of 375a and 377a as well as their rhenium derivatives 375 (Re-aspirin) and 377 (Re-indomethacin) against Ishikawa and HEC-1A cells. Both cell lines responded to the rhenium derivatives, for which the biologically active concentrations were significantly lower than in the case of the original NSAID counterparts. Moreover, the rhenium derivatives showed higher inhibitory activity during prolonged incubation, which was not observed with NSAIDs. These effects were found to be not only time-dependent but also cell line-dependent, as different concentrations were required for 50% inhibition of Ishikawa and HEC-1A viability. The nature of the endometrial cancer cell lines may be the reason for these differences. According to Bokhman’s classification [39], based on different etiopathological, genetic and clinical features, Ishikawa cells belong to type I EC, while HEC-1A cells are representatives of type II. Carcinoma cells of type I had shown high expression of estrogen and progesterone receptors, had low histological differentiation, and are considered typical of early-stage disease. Type II cells represent highly aggressive and invasive carcinoma types, with low levels of expression of estrogen and progesterone receptors and are commonly referred to as typical of the advanced stage of the disease. Moreover, these two cell lines were derived from adenocarcinomas from women at different ages (39-year and 71-year-old for Ishikawa and HEC-1A, respectively), which is associated with the premenopausal and postmenopausal nature of the cells [40]. Ishikawa cell line is a well-differentiated human adenocarcinoma cell line carrying estrogen and progesterone receptors. This cell line is known to produce corticotropin-releasing hormone, placental alkaline phosphatase, and chorionic gonadotropin and is responsive to steroid hormones [41,42]. HEC-1A cancer cells express the wild-type form of the estrogen receptor, and 17-beta-estradiol induces proliferation of these cells. In addition, these cells contain a nonsense mutation in the *hpms2* and a splicing mutation in the *hmsh6* genes, which play key roles in DNA’s methyl-directed mismatch repair (MMR) pathway [43,44]. These differences in characteristics are probably the main reason for the different responses of the cells to treatment with NSAIDs versus rhenium-NSAIDs., The better activity of rhenium derivatives compared with the original NSAIDs was also observed in previous studies performed by our research group using HeLa and L929 cell lines. Rhenium-aspirin and rhenium-indomethacin were more active against cancerous HeLa cells than non-cancerous L929 cells with IC_50_ values of 36 µM and 158 µM, respectively. The IC_50_ values for aspirin and indomethacin against HeLa were much more higher (2819 µM and 326 µM, respectively) [16]. 

The proliferation assay showed higher sensitivity of HEC-1A cells to original NSAIDs and their rhenium derivatives than Ishikawa cells. In this case, HEC-1A cells were affected by both tested rhenium compounds already after the 24-h incubation period, while Ishikawa cells proved susceptible to 377 and 377a. For both cancer cell lines, the anti-proliferative effect increased after 72-h incubation with tested NSAIDs and their rhenium derivatives. The proliferation rate of cells grown under normal conditions varies and depends on the type of cancer. In the case of Ishikawa cells, their proliferation rate is faster than that of HEC-1A cells and increases with the number of passages [40,45]. That may be a possible reason for the lower inhibitory effect of the tested compounds against Ishikawa cells proliferation. Changes in proliferation rate were studied not only at IC_50_ but also at concentrations equal to ½ IC_50_ and ¼ IC_50_ (Table 3 and Table 4). For the lower concentrations, only 375a showed a significant anti-proliferative effect after 24-h incubation, where ½ IC_50_ and ¼ IC_50_ of this compound (* = *p *< 0.05) inhibited cell proliferation of Ishikawa and HEC-1A cells, respectively. This observation is consistent with previous reports showing that low doses of NSAIDs, especially aspirin, provided good survival outcomes for women with endometrial cancer [46,47]. Cisplatin (CisPt) is a good candidate for comparison because of its widespread use as an anti-EC drug as a so-called first-line chemotherapeutic agent, which is administered to patients in single doses as well as in combination chemotherapy (at a dose and regimen of 50–100 mg per m^2^ of a body surface area, every 3 weeks) [48]. CisPt was found to inhibit EC cell proliferation by increasing the expression of semaphorin 3B (SEMA3B), which has suppressive effects on cancer cell proliferation and angiogenesis. As our results showed, after 24-h incubation, CisPt had a significant (* = *p* < 0.05) anti-proliferative effect only on Ishikawa endometrial cancer cells but not on HEC-1A. That could be explained by the overexpression of *BCL2*, which is associated with the resistance of HEC-1A cancer cells to CisPt and the longer incubation time required to transpose the drug across the cell membrane, as shown by Rouette et al. [49]. 

It has already been shown that NSAIDs are associated with an increase in the production of reactive oxygen species (ROS). This mechanism of action of NSAIDs is commonly suggested to initiate side effects, particularly within the gastroenterological tract. However, this mechanism also may be used to destroy cancer cells by inhibiting cell growth and metabolism and disrupting the signaling pathways. Increased levels of ROS induce disruption of cellular homeostasis, leading to cellular damage and initiation of apoptosis [50,51,52]. Therefore, the induction of high ROS levels has been suggested by our group as a positive mechanism of rhenium-NSAIDs activity against previously studied cells [16]. In this study, this effect was evident only in HeLa cancer cells, not in non-neoplastic L-929 fibroblast cells. It was observed that cancer cells have a significantly higher ability to accumulate rhenium derivatives in mitochondria, which could also be related to an increase in mitochondria-specific ROS. As shown in the present study, there was increased production of cytosolic ROS in Ishikawa cells following treatment with both NSAIDs and their rhenium derivatives. Moreover, both NSAIDs and their rhenium derivatives, had a significant impact on ROS levels in HEC-1A, but at ½ IC_50_ concentrations, which may be related to better penetration of compounds across cell membranes. Furthermore, the ½ IC_50_ concentrations of rhenium derivatives **375** and **377** were lower than ½ IC_50_ of original NSAIDS **375a** and **377a**, corresponding to 197.03 µM and 729.65 µM and to 13,627.9 µM and 2744.65 µM, respectively. These results indicate a more effective induction of ROS by rhenium compounds than by their original NSAID counterparts because of the higher activity of the former at lower concentrations than the latter. The result of measuring mitochondrial-specific ROS levels did not match the effects of the agents on cytosolic ROS levels. In this case, the highest level (which is considered to be a 100% increase in ROS amount) for both cell lines appeared to be associated with 377. In the case of Ishikawa cells, this effect was additionally evident for 375, but not to the same extent as for 377. That may be related to the cell-dependent mode of interaction of the compounds with mitochondria. It should additionally be noted that the incubation time of the compounds with cells was only two hours, which is related to the limitations of an assay.

The investigated rhenium derivatives have potential as agents against uterine-related cancers, but the study of the biological activity of these compounds should be further pursued. The next steps should include investigating the mechanisms of rhenium-NSAIDs impact at deeper molecular levels.

## 4. Materials and Methods

### 4.1. NSAIDs and Rhenium-NSAIDs Complexes

NSAIDs and their complexes with rhenium were synthesized in the Department of Organic Chemistry, Faculty of Chemistry, University of Lodz, by K. Kowalski research group [16]. Compounds tested in this study are listed in Table 5.

### 4.2. Human Cell Culture

Adherent human endometrial cancer cell lines Ishikawa (ECACC 99040201) and HEC-1-A (ATTC^®^ HTB-112) were obtained from the European Collection of Authenticated Cell Cultures (ECACC, Porton Down, UK) and the American Type Culture Collection (ATCC^®^, Manassas, VA, USA), respectively. Both cell lines were grown in DMEM-F12 medium with a stable *L*-glutamine, 15 mM HEPES, and supplemented with 10% Ultra-low Endotoxin Fetal Bovine Serum (FBS) (Biowest, Nuaille, France), and 100 U/mL of penicillin and 100 mg/mL of streptomycin (Biowest, Nuaille, France), under standard conditions (37 °C, 5% CO_2_). Cells were passaged every 5–6 days; the growth medium was changed every 2–3 days after a passage. For microplate assays, cells were seeded at the optimal density of 10,000 cells/well.

### 4.3. Cell Viability Assay

The effects of individual NSAIDs and their rhenium derivatives on the endometrial cancer cell viability were evaluated using the MTT (3-(4,5-dimethylthiazol-2-yl)-2,5-diphenyltetrazolium bromide) assay. Ishikawa and HEC-1A cells were initially seeded into 96-well microplates at optimal density (10,000 cells/well) and cultured for the next 24 h under standard conditions. After pre-cultivation, cells were treated with compounds that were added in two-fold dilutions ranging from 10 to 500 µM. All stock solutions of the tested compounds were prepared in DMSO. 1% DMSO was used as a negative control because the concentration of the solvent under experimental conditions never exceeded this value; cisplatin, a well-known anticancer drug was used as a positive control and dissolved in 150 mM saline. Cells were incubated in the presence of the test compounds for 24 and 72 h. After incubation, MTT was dissolved in a culture medium and added to the cells in the final concentration of 0.5 mg/mL. The plates were then incubated for 2 h under standard conditions. Formazan crystals formed during the MTT reaction were solubilized in 150 µL of DMSO. Absorbance was measured at 570 nm using a SpectraMax i3 microplate reader (Molecular Devices, Silicon Valley, CA, USA). The assay was performed in three independent experiments. The IC_50_ (compound concentration inhibiting cell proliferation by 50%) values were determined using GraphPad Prism 7 Software.

### 4.4. BrdU Assay 

The effects of the compounds on cell proliferation were examined using a BrdU cell proliferation assay kit (Biovision, Milpitas, CA, USA). BrdU (5-bromo-2-deoxyuridine) is a pyrimidine analog that is incorporated into the newly synthesized DNA of proliferating cells by replacing thymine. The incorporated BrdU was quantified using mouse anti-BrdU primary antibodies, HRP-linked anti-mouse IgG antibodies, and 3,3′,5,5′-tetramethylbenzidine (TMB) as an HRP substrate, according to the manufacturer’s instructions. Endometrial cancer cells were seeded into 96-well plates at optimal density and precultured overnight under standard conditions. Cells were then treated with the selected compounds at IC_50_, ½ IC_50_, and ¼ IC_50_ concentrations established previously in the MTT assay and incubated under standard conditions for 24 and 72 h. Absorbance was measured at 450 nm using a SpectraMax i3 plate reader (Molecular Devices, Silicon Valley, CA, USA). The assay was performed in three independent experiments. The results were calculated using GraphPad Prism 7 software.

### 4.5. Cellular ROS Level Measurement

Reactive oxygen species (ROS) levels were measured using H_2_DCFDA (ThermoFisher) fluorescent dye. Fluorescence measurement is made by converting H_2_DCF-DA to 2′,7′-dichlorofluorescein (DCF), the amount of which is directly dependent on the amount of ROS produced in the cells. Thus, ROS detection was performed by measuring the fluorescence of H_2_DCF-DA-treated Ishikawa and HEC-1A cancer cells at Ex/Em = 495/520 nm. Ishikawa and HEC-1A cells were seeded into black bottom 96-well cell culture plates. Once cells reached approximately 80–90% confluence, 100 µL of DMEM-F12 medium w/o FBS supplemented with H_2_DCF-DA dye at a final concentration of 5 µM was added. Cells were then incubated for 40 min under standard conditions, protected from light. After incubation, cells were washed twice with DPBS, treated with the selected compound at IC_50_ and ½ IC_50_ concentrations, obtained in the MTT assay, and then incubated for 0.5, 1, and 3 h. After each time point, cells were washed twice with DPBS and suspended in 100 µL of DPBS. Fluorescence was measured at Ex/Em = 495/520 nm using a SpectraMax i3 plate reader (Molecular Devices, Silicon Valley, CA, USA). The assay was performed in three independent experiments, and the results were calculated using GraphPad Prism 7 Software.

### 4.6. Mitochondrial ROS Level Measurement

The assay was performed using the MitoSOX Red dye (Invitrogen, Waltham, MA, USA). When incorporated into mitochondria, this dye is rapidly oxidized by peroxide and converted into the fluorescent product 2-hydroxymitoethidium. Initially, cells were seeded into a black bottom 96-well plate. Once approximately 80–90% cell confluence was achieved, the culture medium was replaced with the medium w/o FBS containing 5 µM MitoSOX Red dye and incubated for 10 min under standard conditions. After incubation, cells were washed twice with DPBS and treated with NSAIDs and rhenium-NSAIDs at IC_50_ concentrations. 1% H_2_O_2_ and cisplatin were used as controls. Cells were incubated with the compounds for 2 h under standard conditions, protected from light. Cells were then washed twice with DPBS and resuspended in 100 µL of DPBS. Fluorescence was measured at Ex/Em = 400/595 nm using SpectraMax i3 plate reader (Molecular Devices, Silicon Valley, CA, USA). The assay was performed in three independent replicates and the results were calculated using GraphPad Prism 7 software.

## Figures and Tables

**Figure 1 ijms-23-11568-f001:**
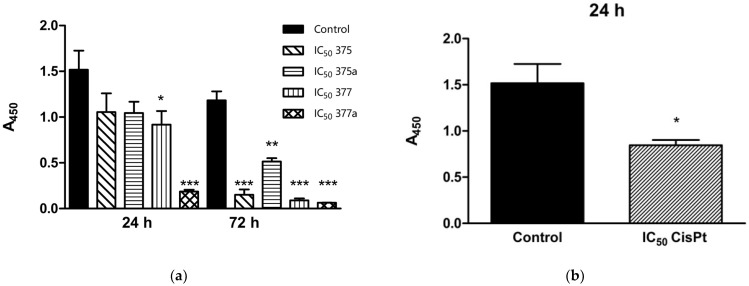
Changes in proliferation status of Ishikawa endometrial cancer cells after 24- and 72-h incubation with **375** (rhenium aspirin), **375a** (aspirin), **377** (rhenium indomethacin) and **377a** (indomethacin) at IC_50_ concentrations (**a**). The assay was performed using the BrdU proliferation kit (Biovision). Cisplatin (CisPt) was used as a positive control (**b**). Data were obtained by measuring absorbance at 450 nm in three independent experiments. Results were statistically compared with the negative control (untreated cells with 1% DMSO dissolved in cell culture medium) using a Two-way ANOVA with Bonferroni multiple comparison test; * = *p* < 0.05, ** = *p* < 0.01, *** = *p* < 0.001.

**Figure 2 ijms-23-11568-f002:**
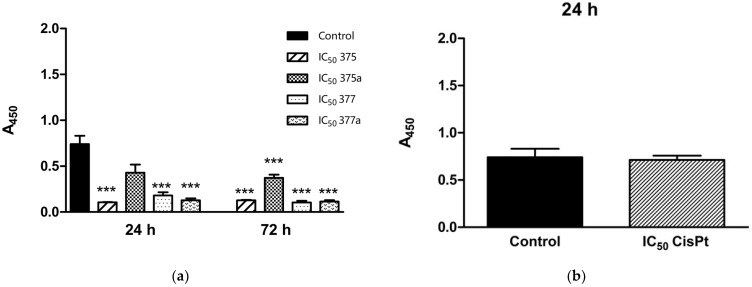
Changes in proliferation status of HEC-1A endometrial cancer cells after 24- and 72-h incubation with **375** (rhenium aspirin), **375a** (aspirin), **377** (rhenium indomethacin) and **377a** (indomethacin) at IC_50_ concentrations (**a**). The assay was performed using the BrdU proliferation kit (Biovision). Cisplatin (CisPt) was used as a positive control (**b**). Data were obtained by measuring absorbance at 450 nm in three independent experiments. Results were statistically compared with the negative control (untreated cells with 1% DMSO dissolved in cell culture medium) using a Two-way ANOVA with Bonferroni multiple comparison test; *** = *p* < 0.001.

**Figure 3 ijms-23-11568-f003:**
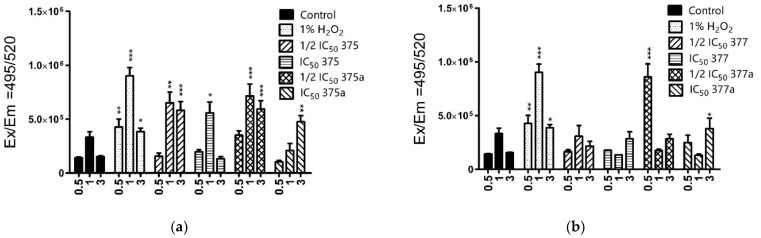
Cytosolic levels of reactive oxygen species in Ishikawa cells after 0.5, 1 and 3-h treatment with **375** (rhenium aspirin), **375a** (aspirin) (**a**), **377** (rhenium indomethacin) and **377a** (indomethacin) (**b**). Cells were preincubated with H_2_DCF-DA dye and then treated with the tested drugs. Data were obtained by measuring the fluorescence at Ex/Em = 495/520 nm. 1% H_2_O_2_ was used as a positive control. Results were statistically compared with the negative control (untreated cells with 1% DMSO dissolved in cell culture medium). Two-way ANOVA with Bonferroni multiple comparison test was used for statistical analyses; * = *p* < 0.05, ** = *p* < 0.01, *** = *p* < 0.001.

**Figure 4 ijms-23-11568-f004:**
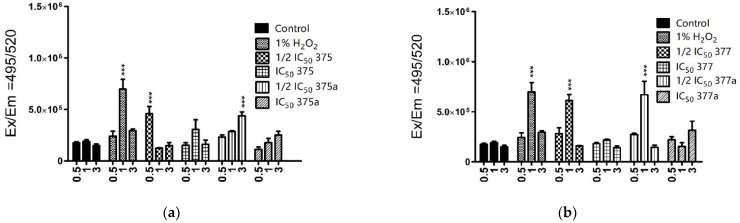
Cytosolic levels of reactive oxygen species in HEC-1A cells after 0.5, 1 and 3 h-treatment with **375** (rhenium aspirin), **375a** (aspirin) (**a**), **377** (rhenium indomethacin) and **377a** (indomethacin) (**b**). Cells were preincubated with H2DCF-DA dye and then treated with the tested drugs. Data were obtained by measuring the fluorescence at Ex/Em = 495/520 nm. 1% H_2_O_2_ was used as a positive control. Results were statistically compared with the negative control (untreated cells with 1% DMSO dissolved in cell culture medium). Two-way ANOVA with Bonferroni multiple comparison test was used for statistical analyses; *** = *p* < 0.001.

**Figure 5 ijms-23-11568-f005:**
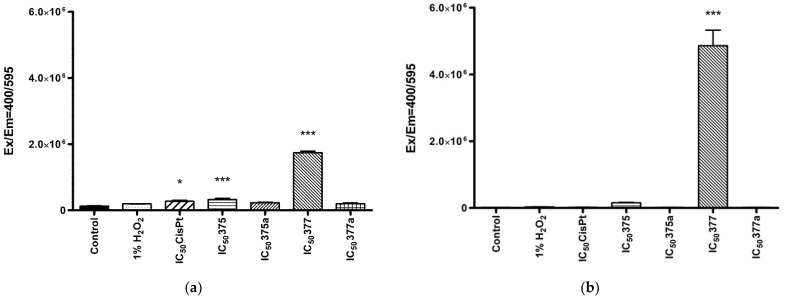
(**a**). Mitochondrial levels of reactive oxygen species in Ishikawa (**a**) and HEC-1A (**b**) cells after 2-h treatment with **375** (rhenium aspirin), **375a** (aspirin), **377** (rhenium indomethacin) and **377a** (indomethacin). Cells were preincubated with MitoSOX Red dye and then treated with the tested drugs. Data were obtained by measuring the fluorescence at Ex/Em = 400/595 nm. 1% H_2_O_2_ was used as a positive control. Results were statistically compared with the negative control (untreated cells with 1% DMSO dissolved in cell culture medium). Two-way ANOVA with Bonferroni multiple comparison test was used for statistical analyses; * = *p* < 0.05, *** = *p* < 0.001.

**Table 1 ijms-23-11568-t001:** Effect of studied NSAIDs and their rhenium derivatives on the viability of Ishikawa cell line after 24- and 72-h treatment; the results were expressed as the mean IC_50_ (µM; µg/mL) from three independent experiments; CisPt–cisplatin.

**24 h**									
**IC_50_**	**373**	**373a**	**375**	**375a**	**376**	**376a**	**377**	**377a**	**CisPt**
µg/mL	na *	na	11.83	180.6	na	na	18.45	117.9	4.072
µM	na	na	188.06	10,024.42	na	na	228.6	3295.3	13.57
**72 h**									
**IC_50_**	**373**	**373a**	**375**	**375a**	**376**	**376a**	**377**	**377a**	**CisPt**
µg/mL	na	na	9.208	313.4	na	na	8.695	169.5	12.31
µM	na	na	146.3	17,395.6	na	na	107.7	4737.4	41.03

* na—no activity against studied cells; **373a**—naproxen; **373**—rhenium naproxen; **375a**—aspirin; **375**—rhenium aspirin; **376a**—ibuprofen; **376**—rhenium ibuprofen; **377a**—indomethacin; **377**—rhenium indomethacin.

**Table 2 ijms-23-11568-t002:** Effect of studied NSAIDs and their rhenium derivatives on the viability of HEC-1A cell line after 24- and 72-h treatment; the results were expressed as the mean IC_50_ (µM; µg/mL) from three independent experiments; CisPt–cisplatin.

**24 h**									
**IC_50_**	**373**	**373a**	**375**	**375a**	**376**	**376a**	**377**	**377a**	**CisPt**
µg/mL	na *	na	24.81	490.5	na	na	117.8	196.4	6.016
µM	na	na	394.06	27,255.8	na	na	1459.3	5489.3	20.05
**72 h**									
**IC_50_**	**373**	**373a**	**375**	**375a**	**376**	**376a**	**377**	**377a**	**CisPt**
µg/mL	na	na	7.702	309.3	na	na	11.18	228.1	3.083
µM	na	na	112.4	17,168	na	na	138.5	6375.2	10.27

* na—no activity against studied cells; **373a**—naproxen; **373**—rhenium naproxen; **375a**—aspirin; **375**—rhenium aspirin; **376a**—ibuprofen; **376**—rhenium ibuprofen; **377a**—indomethacin; **377**—rhenium indomethacin.

**Table 3 ijms-23-11568-t003:** Effect of **375**, **375a**, **377**, and **377a** at IC_50_, ½ IC_50_, and ¼ IC_50_ concentrations on Ishikawa cells proliferation. Results are presented for 24- and 72-h incubation periods. One-way ANOVA with Bonferroni’s comparison test was chosen for statistical analysis.

	Compound	*p* < 0.05	SD	Compound	*p* < 0.05	SD
24 h	375 at ½ IC_50_	no *	±0.005102	375a at ½ IC_50_	*	±0.00035
	375 at ¼ IC_50_	no	±0.0002	375a at ¼ IC_50_	no	±0.00015
	377 at ½ IC_50_	***	±0.0003	377a at ½ IC_50_	***	±0.02975
	377 at ¼ IC_50_	***	±0.1768	377a at ¼ IC_50_	***	±0.1933
72 h	375 at ½ IC_50_	***	±0.0308	375a at ½ IC_50_	***	±0.1401
	375 at ¼ IC_50_	***	±0.01290	375a at ¼ IC_50_	***	±0.01455
	377 at ½ IC_50_	***	±0.0432	377a at ½ IC_50_	***	±0.07570
	377 at ¼ IC_50_	***	±0.0914	377a at ¼ IC_50_	***	±0.1038

* no—no significant changes; **373a**—naproxen; **373**—rhenium naproxen; **375a**—aspirin; **375**—rhenium aspirin; **376a**—ibuprofen; **376**—rhenium ibuprofen; **377a**—indomethacin; **377**—rhenium indomethacin; SD—standard deviation; * = *p* < 0.05, *** = *p*< 0.001.

**Table 4 ijms-23-11568-t004:** Effect of **375**, **375a**, **377**, and **377a** at IC_50_, ½ IC_50_, and ¼ IC_50_ concentrations on HEC-1A cells proliferation. Results are presented for 24- and 72-h incubation periods. One-way ANOVA with Bonferroni’s comparison test was chosen for statistical analysis.

	Compound	*p* < 0.05	SD	Compound	*p* < 0.05	SD
24 h	375 at ½ IC_50_	*	±0.0600	375a at ½ IC_50_	no *	±0.04030
	375 at ¼ IC_50_	no	±0.3835	375a at ¼ IC_50_	*	±0.4275
	377 at ½ IC_50_	no	±0.1122	377a at ½ IC_50_	no	±0.1232
	377 at ¼ IC_50_	no	±0.1461	377a at ¼ IC_50_	no	±0.1284
72 h	375 at ½ IC_50_	***	±0.0115	375a at ½ IC_50_	***	±0.0886
	375 at ¼ IC_50_	***	±0.1675	375a at ¼ IC_50_	***	±0.1061
	377 at ½ IC_50_	***	±0.04905	377a at ½ IC_50_	***	±0.05215
	377 at ¼ IC_50_	***	±0.0171	377a at ¼ IC_50_	**	±0.3529

* no—no significant changes; **373a**—naproxen; **373**—rhenium naproxen; **375a**—aspirin; **375**—rhenium aspirin; **376a**—ibuprofen; **376**—rhenium ibuprofen; **377a**—indomethacin; **377**—rhenium indomethacin; SD—standard deviation; * = *p* < 0.05, ** = *p*< 0.01, *** = *p*< 0.001.

**Table 5 ijms-23-11568-t005:** NSAIDs and their corresponding rhenium complexes used in cell viability assay of Ishikawa and HEC-1A cell lines.

NSAIDs and Rhenium-NSAIDs Agents		
**Structure**	Name/Formula	Molecular Weight [g/mol]
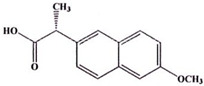	Naproxen (373a) C_14_H_14_O_3_	230.26
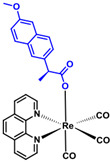	Rhenium-naproxen (373) C_29_H_21_N_2_O_6_Re_1_	679.69
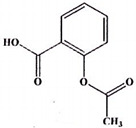	Aspirin (375a) C_9_H_8_O_4_	180.16
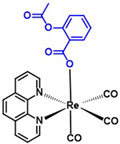	Rhenium-aspirin (375) C_24_H_15_N_2_O_7_Re_1_	629.59
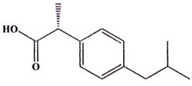	Ibuprofen (376a) C_14_H_22_O_2_	206.28
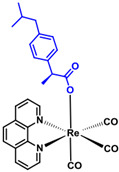	Rhenium-ibuprofen (376) C_28_H_25_N_2_O_5_Re_1_	655.72
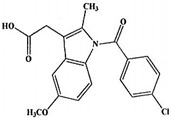	Indomethacin (377a) C_19_H_16_Cl_1_N_1_O_4_	357.79
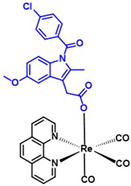	Rhenium-indomethacin (377); C_34_H_24_Cl_1_N_3_O_7_Re_1_	807.22

## Data Availability

Not applicable.

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
