# Peer review of "Biological Activity of fac-[Re(CO)3(phen)(aspirin)], fac-[Re(CO)3(phen)(indomethacin)] and Their Original Counterparts against Ishikawa and HEC-1A Endometrial Cancer Cells"

_ijms, 2022, doi:10.3390/ijms231911568_

Round 1

Reviewer 2 Report

This manuscript by Kuźmycz et al. describes “Biological activity of fac-[Re(CO)3(phen)(aspirin)] and fac-[Re(CO)3(phen)(indomethacin)] against Ishikawa and HEC-1A endometrial cancer cells”. This is a nice follow-up study against different cancer cells than the previous report in the New Journal of Chemistry 2019, 43(2), 573–583. The manuscript is well written. These NSAIDs-rhenium derivatives displayed improved activity in compared to parent NSAIDs. I would recommend this manuscript to be published after minor comments as below

1.     Page 1, line 41, introduction section, please cite references for several research focused on the role of inflammatory processes in the development of gynecologic cancers.

2.     Page 4, section 2.3.1 and 2.3.2, write names of the dyes (in the text) used for cytosolic and mitochondrial ROS level measurement. Authors can discuss about quantification of ROS generated in comparison to control by compound 377.  What was the reason authors incubated cells with compounds just for two hours to measure mitochondrial ROS?   

3. Page 10 and 11, molecular weight should have dot instead of comma and at some places H2O2 can be corrected to H2O2. Manuscript can be checked such minor errors and should be corrected in final version.

Reviewer 3 Report

ijms-1924541-peer-review-v1

The present paper present interesting results of the application of modified NSAID with potential anticancer properties. Work was well planed and presented and, in my opinion, deserve to be accepted for publication, however, some adjustments and corrections needs to be taken into consideration by the authors.

Ln67: Even if codes for each compound was stated in the text (Ln64-65), They still need to be included in the table description, or as note after the table. Table/s need to self explaining. Same for Table 2.

In my opinion, codes for the different compounds do not need to be in bold.

Table 3: You can replace “std. deviation” with “SD”

Fig 3 and 4 can be combined as Fig.3a and 3b.

Figure 5 and 6 can be combined as fig. 5a and 5b.

Ln223. At the end of the first sentence of the discussion, a reference needs to be provided.

Ln342-343: The sentence is confusing. In present way it is much more as a results statement, and not as material and methods. Please, correct it.

It is correct (according to the Journal recommendations) to have materials and methods after Results and Discussion?

Ln344-345: On Ln 342-343 you stated that only 4 were active, but in the title was stated that all were active. What is correct?

Ln346: There was no antibiotics added to the growth medium?

Round 2

Reviewer 1 Report

Thank you for making corrections. I think the research is publishable. However, in the future, I suggest that the ROS studies be better refined so that both positive and negative controls are clearly defined.